# Gestational Weight Gain—Re-Examining the Current Paradigm

**DOI:** 10.3390/nu12082314

**Published:** 2020-08-01

**Authors:** Jennie Louise, Andrea R. Deussen, Jodie M. Dodd

**Affiliations:** 1The Robinson Research Institute and Department of Women’s and Children’s Health, The University of Adelaide, Adelaide, SA 5006, Australia; jennie.louise@adelaide.edu.au (J.L.); andrea.deussen@adelaide.edu.au (A.R.D.); 2Women’s and Babies Division, Department of Perinatal Medicine, The Women’s and Children’s Hospital, Adelaide, SA 5006, Australia

**Keywords:** gestational weight gain, pregnancy dietary and lifestyle intervention, obesity, causal

## Abstract

Our aim was to investigate the underlying assumptions of the current gestational weight gain (GWG) paradigm, specifically that—(1) GWG is modifiable through diet and physical activity; (2) optimal GWG and risk of excess GWG, vary by pre-pregnancy body mass index (BMI) category and (3) the association between GWG and adverse pregnancy outcomes is causal. Using data from three large, harmonized randomized controlled trials (RCTs) of interventions to limit GWG and improve pregnancy outcomes and with appropriate regression models, we investigated the link between diet and physical activity and GWG; the relationships between pre-pregnancy BMI, GWG and birth weight z-score; and the evidence for a causal relationship between GWG and pregnancy outcomes. We found little evidence that diet and physical activity in pregnancy affected GWG and that the observed relationships between GWG and adverse pregnancy outcomes are causal in nature. Further, while there is evidence that optimal GWG may be lower for women with higher BMI, target ranges defined by BMI categories do not accurately reflect risk of adverse outcomes. Our findings cast doubt upon current advice regarding GWG, particularly for overweight and obese women and suggest that a change in focus is warranted.

## 1. Introduction

High gestational weight gain (GWG) has been identified as a risk factor for the occurrence of adverse maternal and infant outcomes during pregnancy and childbirth [1,2,3] and for increased postpartum weight retention [4]. Furthermore, high GWG is strongly associated with high infant birth weight and independently associated with an increased risk of child obesity in the offspring [5,6]. This potentially creates a vicious cycle in which the intergenerational effects of obesity are perpetuated [7].

The US-based Institute of Medicine (IOM) has summarized considerable observational literature relating to GWG [8,9]. Recommendations to minimize adverse pregnancy outcomes advise weight gain between 11.5–16.0 kg for women with a body mass index (BMI) of 18.5–24.9 kg/m^2^, categorized as normal, 7.0–11.5 kg for women with a BMI of 25.0–29.9 kg/m^2^, categorized as overweight and 5.0–9.0 kg for women with a BMI of 30 kg/m^2^ or more categorized as obese [9]. These ranges were identified as those in which the risk of adverse maternal and newborn outcomes was lowest, the composite including the birth of an infant small (SGA) or large (LGA) for gestational age, caesarean section, preterm birth and postpartum weight retention [9]. Subsequent reports confirm the association between ‘excess’ or GWG above the optimal range and increased risk of adverse pregnancy outcomes, including LGA, caesarean birth and preterm birth [10,11].

Many clinical guidelines therefore advocate that pregnant women can lower their risk of adverse outcomes by ensuring that their GWG falls within the optimal range for their pre-pregnancy BMI and further suggest that this can be achieved by adopting a healthy diet and physical activity [12,13,14]. Such advice assumes that the observed associations between GWG and adverse outcomes are causal and that GWG is modifiable through diet and physical activity. The use of optimal ranges based on pre-pregnancy BMI and the focus on overweight and obese women as specific target groups for interventions designed to limit GWG, further assumes that optimal GWG varies by pre-pregnancy BMI category and that overweight and obese women are at particular risk of excess GWG.

However, the accumulated evidence from numerous randomized trials (RCTs) of antenatal dietary and lifestyle interventions conducted over the past decade has not supported all of these assumptions. These trials were implemented in the expectation that an effective intervention would reduce excessive GWG and thereby improve pregnancy outcomes, with many specifically targeted to women with overweight and obesity as an identified high-risk group. Overall, little effect on GWG or impact on maternal or infant outcomes has been demonstrated [15,16].

Our aim was to investigate the underlying assumptions of the current GWG paradigm using data from our suite of harmonized RCTs (the LIMIT [17], GROW [18] and OPTIMISE [19] trials). Our specific research questions were:Is GWG modifiable through diet and physical activity?Does optimal GWG and risk of excess GWG, vary by pre-pregnancy BMI category?Is the association between GWG and adverse pregnancy outcomes causal?

## 2. Materials and Methods 

Our group has previously conducted three large RCTs of antenatal interventions to limit GWG and improve pregnancy outcomes—the LIMIT trial (2212 randomized participants) of an antenatal diet and lifestyle intervention in women with BMI ≥ 25.0 kg/m^2^ [17]; the GRoW trial (524 randomized participants) of metformin in addition to diet and lifestyle advice in women with BMI ≥ 25.0 kg/m^2^ [18]; and the OPTIMISE trial (641 randomized participants) of an antenatal diet and lifestyle intervention in women with BMI 18.5–24.9 kg/m^2^ [19]. The data from these three trials—with the same dietary and lifestyle intervention implemented, consistent data collection and outcomes and participants recruited from the same population within a ten year time period—provide a unique opportunity to investigate the existence and nature, of pathways between pre-pregnancy BMI, antenatal diet and physical activity, GWG and maternal and infant pregnancy outcomes. 

In brief, the LIMIT Trial randomized 2212 women to either Lifestyle Advice or Standard Care [17]. Diet quality and physical activity were improved in participants receiving the intervention [20], although there was no significant difference in total GWG (mean difference −0.04 kg, 95% CI −0.55, 0.48 kg) or in the risk of excess GWG (RR 0.99, 95% CI 0.89, 1.10) [17]. The intervention group did have a significantly lower rate of birth weight > 4 kg (RR 0.82, 95% CI: 0.68–0.99) but no significant effects were observed for other clinical outcomes [17,21].

GRoW was a randomized, double-blind, placebo-controlled trial of metformin in addition to a diet and lifestyle intervention to limit GWG and improve pregnancy outcomes in women with BMI ≥ 25.0 kg/m^2^, involving a total of 524 women [18]. There was weak evidence to suggest that Lifestyle Advice Plus Metformin reduced average weekly GWG by 0.08 kg (95% CI: 0.02 kg, 0.14 kg) in the intervention group and these participants were also more likely to have GWG below the recommended range (RR 1.46, 95% CI: 1.10, 1.94) [18]. However the evidence for a reduction in total GWG was weak (mean difference −1.18 kg, 95% CI −2.37, 0.01 kg) and there was no significant difference in risk of excess GWG (RR 0.84, 95% CI: 0.65, 1.09) or in maternal or infant outcomes [18].

The OPTIMISE RCT in women with a ‘normal’ BMI (18.5–24.9 kg/m^2^) involved 641 women randomized either to Lifestyle Advice or Standard Care [19]. While the intervention improved diet quality, there was no evidence for an effect on total GWG (mean difference −0.37, 95% CI −0.97, 0.23) and only weak evidence of a reduction in risk of GWG above guidelines (RR 0.58, 95% CI: 0.32, 1.04). There was likewise no evidence of an effect on clinical maternal and infant outcomes [19].

The findings of each trial have been reported in detail and have been summarized in Table 1.

Combined, the data from these studies allowed us to investigate a range of questions relating to GWG, pre-pregnancy BMI category and maternal and infant outcomes.

### 2.1. Is Gestational Weight Gain Modifiable through Antenatal Diet and Physical Activity?

In order for dietary and physical activity modifications to effect GWG, it must be true that differences in diet and physical activity cause differences in GWG. To investigate the relationship between diet, physical activity and total GWG, we used data from the Standard Care (control) groups of the LIMIT (women with BMI ≥ 25.0 kg/m^2^) [17,20] and OPTIMISE trials (women with BMI 18.5–24.9 kg/m^2^) [19]. Dietary intake data were derived from the Harvard Semi-Structured Food Frequency Questionnaire, completed at trial entry, 28 weeks’ gestation and 36 weeks’ gestation and included total energy intake (kJ), intake of carbohydrate, fiber, fat, protein and sugars (g) and the Healthy Eating Index (HEI). Physical activity was measured by metabolic equivalent task units (METs) per week, calculated from the Short Questionnaire to Assess Health-enhancing Physical Activity, completed at the same time and covering the same periods, as the food frequency questionnaires.

The association between dietary characteristics and total GWG was investigated using linear regression models with total GWG as the outcome and dietary intakes as the predictors. Models were also adjusted for maternal BMI, parity, maternal age at trial entry and Socio-economic index for areas (SEIFA) Quintile of Index of Relative Socio-Economic Disadvantage (IRSD quintile) [22]. Because the relationship between diet or physical activity and GWG may be different between normal BMI and overweight and obese women, we analyzed LIMIT data and OPTIMISE data in separate models, with results presented as the estimated difference in average GWG corresponding to the stipulated increase in intake or activity at each time point. 

### 2.2. Does Optimal GWG and Risk of Excess GWG, Vary by Pre-Pregnancy BMI Category?

In order to investigate whether GWG ranges based on BMI category were likely to be an adequate representation of risk of adverse pregnancy outcomes and whether women in higher BMI categories were indeed at higher risk of “excess” GWG, we performed a range of analyses to characterize the relationships between pre-pregnancy BMI, GWG and birth weight z-score. We chose birth weight z-score as the outcome of interest as it is continuous, standardized to gestational age at birth and represents the outcome for which evidence of an association with GWG is strongest [9]. Moreover, it can be assumed that an increase in mean birth weight z-score implies both a lower risk of SGA and a higher risk of LGA.

Firstly, we used descriptive analysis and regression modelling to characterize the association between pre-pregnancy BMI and GWG, using data from the Standard Care groups of the LIMIT and OPTIMISE studies. We initially used fractional polynomial modelling [23,24] to allow for nonlinearity in the relationship but as there was no evidence such polynomial terms were required, we then used standard linear regression to model the relationship between pre-pregnancy BMI and total GWG. We then investigated how the risk of “excess” GWG was associated with BMI categories and with BMI as a continuous phenomenon. To do this, we calculated the distance between each participants’ actual BMI and the lower value for their BMI category and investigated how risk of excess GWG was related to BMI category, distance from the lower boundary of the category and the interaction between these, using robust log Poisson regression. The model was also adjusted for parity, age, smoking status and SEIFA IRSD quintile [22]. We calculated marginal estimates for the proportion of women with excess GWG in each BMI category and the Relative Risk of excess GWG corresponding to an increase of 1 kg/m^2^ over the cut-off in each BMI category.

Secondly, we performed linear regression analyses to determine the effects of pre-pregnancy BMI, GWG and their interaction, on birth weight z-score. For these analyses we used data from both the Standard Care and Intervention groups of LIMIT and OPTIMISE, in order to maximize statistical power to detect interaction effects and because there was no reason to believe that the intervention altered the causal relationships under investigation. To evaluate the relationship between pre-pregnancy BMI and GWG, we initially investigated possible nonlinearity using fractional polynomials. We also fitted an initial model containing a 3-way interaction between BMI category, distance from BMI category cutoff and total GWG, in order to examine the relative contributions of BMI category and BMI as a continuous phenomenon. Having determined that neither nonlinear terms nor 3-way interaction terms were required, we fitted a linear regression model with a two-way interaction between BMI (continuous) and total GWG; the model was also adjusted for parity, age, smoking status and IRSD quintile [22]. From this model, we estimated the mean birth weight z-score at a range of BMI values for a fixed GWG (of 10 kg) and the effect of an increase of 1 kg in total GWG at each of these BMI values. 

### 2.3. Is the Association between GWG and Adverse Pregnancy Outcomes Causal?

To investigate whether the association between GWG and adverse pregnancy outcomes is causal, such that interventions that affect GWG could thereby affect outcomes, we considered three outcomes strongly associated with ‘excess’ GWG: LGA, Caesarean section and birth weight z-score [15]. Data from the GRoW study were used for this analysis, as (unlike LIMIT and OPTIMISE) there was some weak evidence for an intervention effect on GWG and on caesarean section, thus enabling investigation of the causal question by testing whether changes in GWG caused change in outcomes. As above, LGA and birth weight z-score are the outcomes for which evidence of an association with GWG is strongest; caesarean section was added as another outcome commonly associated with GWG and for which there was some evidence of an intervention effect [15]. Other outcomes reportedly associated with ‘excess’ GWG including preterm birth and pre-eclampsia were not investigated as the number of events were too small to provide adequate statistical power, and/or there was a high risk of bias due to data not being missing at random. We first undertook a descriptive analysis comparing rates of LGA and Caesarean section and mean birth weight z-score, with mean GWG across BMI categorized in 2-kg/m^2^ increments. Then, to examine whether (and to what extent) the effect of maternal BMI was mediated through GWG and whether it modified the effect of GWG, we used regression-based mediation modelling [25] to investigate the total, direct (unmediated) and indirect (mediated) effects of the GRoW intervention on LGA, Caesarean section and birth weight z-score. The models allowed for an interaction between the intervention (Metformin) and the mediator (GWG) and were adjusted for maternal pre-pregnancy BMI, age, parity, smoking status and IRSD quintile [22].

## 3. Results

The baseline characteristics of participants included in these analyses are outlined in Table 2.

### 3.1. Is Gestational Weight Gain Modifiable through Antenatal Diet and Physical Activity?

Results are presented in Table 3 and show the estimated difference in average GWG corresponding to increases in intake or activity in pregnancy. There is little evidence for a relationship between dietary intakes and total GWG or between physical activity levels and GWG, in either overweight or obese (LIMIT) or normal BMI (OPTIMISE) women. While some individual estimates (e.g., total energy intake at 28 weeks in the LIMIT Standard Care group) are statistically significant, there is no consistent pattern of association. These estimates also have not been adjusted for multiple comparisons (and would not remain significant after such adjustment), are of small magnitude and in the ‘wrong’ direction (increased intake associated with a decrease in total GWG). Further analyses investigating associations between diet or physical activity and risk of excess GWG (Appendix A) produced similar findings.

### 3.2. Does Optimal GWG and Risk of Excess GWG, Vary by Pre-Pregnancy BMI Category?

There was a strong negative relationship between pre-pregnancy BMI and total GWG (Figure 1). Each kg/m^2^ increase in BMI is associated with a decrease in total GWG of −0.25 (95% CI: −0.30, −0.21) kg on average. Figure 2 illustrates the continuous relationship of GWG by BMI with the IOM recommended weight gain categories superimposed. Results from models estimating relative risk of excess GWG by BMI category, distance from BMI category boundary and their interaction, are presented in Table 4. Overall, the proportion of women gaining in excess of the recommended GWG range in the overweight and obese BMI categories was significantly higher than that in the normal BMI category, with 18% (95% CI: 0–37%) of women in the normal BMI category, 38% (95% CI: 32–43%) of women in the overweight category and 46% (95% CI: 42–51%) in the obese category, gaining above the recommended range. However, for women in the overweight and obese categories, the relative risk of excess GWG decreased with distance from the BMI category boundary. For women in the overweight category, each kg/m^2^ above 25 was associated with a 12% lower (95% CI: 4–19% lower) risk of excess GWG, while for women in the obese category, each kg/m^2^ above 30 was associated with a 6% (95% CI: 3–8%) lower risk of excess GWG.

Both pre-pregnancy BMI and GWG had a strong positive association with birth weight z-score (Table 5). However, there was little evidence for effect modification, the effect of GWG being quite consistent across all values of pre-pregnancy BMI. Estimated mean birth weight z-score increased with pre-pregnancy BMI. For a fixed GWG of 10 kg, estimated birth weight z-scores corresponding to a pre-pregnancy BMIs of 20, 25, 30 and 35 kg/m^2^ were −0.08 (95% CI −0.15, −0.01), 0.11 (95% CI: 0.06, 0.16), 0.31 (95% CI: 0.27, 0.35) and 0.5 (0.45, 0.56) respectively. At any pre-pregnancy BMI, an increase in total GWG of 1 kg was associated with an increase in birth weight z-score of 0.04 to 0.08.

### 3.3. Is the Association between GWG and Adverse Pregnancy Outcomes Causal?

There was no correspondence between rates of adverse outcomes and birth weight z-score with mean GWG across 2-unit increments of BMI (Appendix A; Appendix A). Appendix A illustrate the relationship between pre-pregnancy BMI and GWG (standardized as number and standard deviation (SD) from the mean) in the control groups from all three studies along with (a) proportions of LGA infants, (b) proportion of caesarean deliveries and (c) birth weight z-score. The proportion of women with excess GWG decreases as BMI increases, whilst proportion of LGA, caesarean birth and birth weight z-score increase.

Results of mediation models investigating the direct and indirect (GWG-mediated) effect of pre-pregnancy BMI on birth weight z-score, LGA and Caesarean section in GRoW participants are reported in Table 6. A significant overall effect of metformin was demonstrated only for Caesarean section but there is no evidence that this effect was mediated via an effect on GWG. The estimated direct effect (through causal pathways other than GWG) is a relative risk of 0.57 (95% CI: 0.37, 0.88), while the estimated indirect effect (mediated via GWG) is 0.99 (0.94, 1.04). By contrast, while there is no strong evidence for an overall effect of metformin on birth weight z-score or LGA, there is some weak evidence of an indirect effect, where the intervention results in a decrease in birth weight z-score of −0.06 (95% CI: −0.14, 0.01) via its effect on GWG.

## 4. Discussion

Our analyses utilized data from our suite of large randomized controlled trials of antenatal lifestyle interventions to investigate a range of research questions relating to pre-pregnancy BMI, GWG and adverse pregnancy outcomes. Our findings suggest that a rethinking of many aspects of the current paradigm regarding GWG is warranted.

Firstly, there is little evidence for a relationship between dietary intake or physical activity and GWG. This is consistent with findings from other studies [26] and suggests that healthy diet and physical activity in pregnancy, however sensible for its own sake, should not be promoted as a method to ensure GWG within the recommended limits.

Secondly, while there is evidence that women in higher BMI categories are more likely to have GWG in excess of the recommended ranges, there is some reason to believe that optimal GWG ranges should not be defined relative to maternal BMI category. Some criticism of the current GWG ranges has already noted that there is only one target range for women with a BMI over 30 kg/m^2^, with suggestions that optimal GWG at much higher BMIs may in fact be lower than that advised by the IOM, extending even to weight loss [27,28]. However, the underlying issue, illustrated in Figure 2, is that of using categories when the underlying phenomenon is continuous; a practice long criticized in the statistical literature both in general [29,30] and in relation to BMI in particular [31]. Because of the sharp changes in recommended ranges at the BMI category boundaries, a woman with BMI at or just above, the boundary will have a very different ‘target’ range to a woman just below it, even though their risk of adverse outcomes is likely to differ very little. For example, a woman with a BMI of 24.9 kg/m^2^ and total GWG of 12 kg has ‘appropriate’ GWG, while for a woman with a BMI of 25.1 kg/m^2^ the same GWG is categorized as ‘excessive.’ Conversely, as we have shown, the risk of ‘excess’ GWG decreases with increased BMI within a category, even though the risks of adverse outcomes continue to increase. Hence, based on the currently defined GWG ranges, a woman with a BMI of 25.0 kg/m^2^ has a higher chance of ‘excess’ GWG than one with a BMI of 29.0 kg/m^2^, even though her overall risk of adverse outcomes at any given GWG value is lower. Overall, as we have shown, a woman’s risk of ‘excess’ GWG does not accurately track her risk of adverse pregnancy outcomes.

It is nevertheless plausible and consistent with our findings, that the range of GWG associated with lowest risk of adverse pregnancy outcomes is lower for women with a higher BMI, since the risk of increased GWG is added to the risk of a higher pre-pregnancy BMI. If average birth weight (relative to gestational age) increases with increasing BMI and also increases with increasing GWG, then risks associated with infants being too small are already much lower in women with higher BMI and increases in GWG serve only to increase the risk associated with infants being too large.

This, however, brings us to our final finding, which is that the evidence for a causal relationship between GWG and pregnancy outcomes is lacking. There was no evidence from our data for the hypothesis that interventions which successfully reduce GWG will thereby improve pregnancy outcomes. The lack of strong effects from intervention studies is admittedly a limitation in attempting to determine whether these effects are mediated via GWG. However, it should also be kept in mind that, while GWG is a relatively easy measure to obtain, it is in fact a composite outcome, reflecting a combination of maternal fat deposition, pregnancy related plasma volume expansion, breast and uterine tissue hypertrophy, extracellular fluid, placental mass, fetal mass and amniotic fluid volume [32]. Furthermore, the relative contribution from each component in any individual woman is difficult to determine. The inclusion of fetal weight (the outcome) in total GWG (the supposed cause) also casts doubt upon any attempt to define the association between the two as causal. Higher fetal weight means that total GWG will necessarily be higher, not just due to the fetal weight itself but also the associated greater placental mass and increased amniotic fluid volume. Even if it were demonstrated that an intervention which reduced GWG also reduced birth weight, without being able to separate out the different components of GWG, we could not be sure that we were not merely measuring the same effect twice.

Strengths of these analyses are that the data presented here were prospectively collected from women participating in three large robust randomized trials [17,18,19] from the same population over a nine year period, where BMI was consistently measured by trained research staff. Outcome data were collected by trained research staff blinded to treatment allocation within the respective studies. Limitations of this work reflect the recruitment of women of predominantly Caucasian ethnicity, however this is reflective of the broader South Australian population demographic distribution. Furthermore, there is an over representation of women of high socio-economic disadvantage, with up to 75% of women declining participation. These factors may limit the external generalizability of our findings and our methods and findings warrant replication in other available birth cohorts.

## 5. Conclusions

Overall, our findings do not provide good evidence that GWG is the appropriate causal mechanism to target for pregnant women who are overweight or obese and may explain the lack of clinical effect observed from antenatal intervention trials. If health outcomes are to be improved, we require a paradigm shift. Continued focus on GWG provides an overly simplistic approach to what are highly complex physiological relationships and is unlikely to be successful. Similarly, a relentless search for the “right” antenatal intervention targeting GWG is unlikely to be successful in view of amassed literature to date.

It is time to redirect efforts and optimize maternal weight prior to conception. To be successful this will require a concerted “whole of system” effort beginning in childhood and adolescence, well before pregnancy is contemplated [33].

## Figures and Tables

**Figure 1 nutrients-12-02314-f001:**
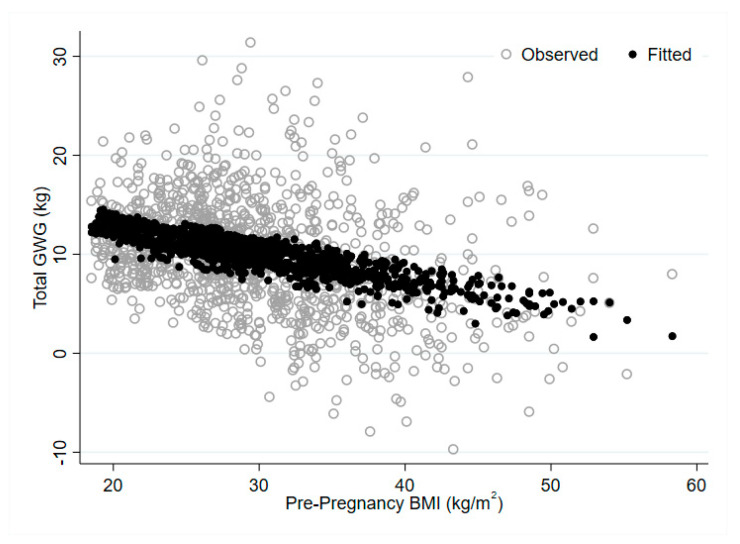
Relationship between pre-pregnancy body mass index (BMI) and gestational weight gain.

**Figure 2 nutrients-12-02314-f002:**
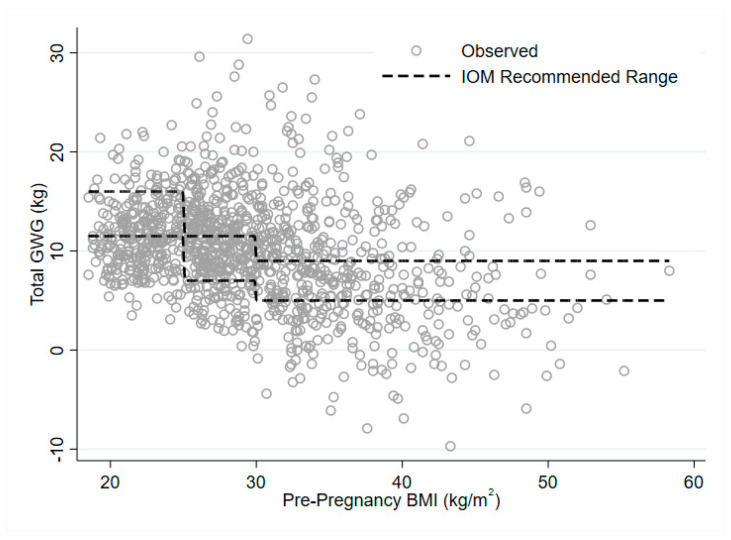
Institute of Medicine (IOM) categories superimposed over gestational weight gain by BMI.

**Table 1 nutrients-12-02314-t001:** Summary of selected findings from LIMIT, GRoW and OPTIMISE.

Outcome	LIMIT	GRoW	OPTIMISE
LGA ^a^	0.90 (0.77, 1.07)	0.87 (0.62, 1.23)	0.88 (0.51, 1.52)
Birth weight > 4 kg ^a^	0.82 (0.68, 0.99)	0.97 (0.65, 1.47)	0.91 (0.54, 1.55)
Birth weight (g) ^b^	−6.90 (−55.47, 41.67)	−13.01 (−106.45, 80.44)	−78.39 (−164.00, 7.22)
Birth weight z-score ^b^	−0.05 (−0.14, 0.03)	−0.06 (−0.24, 0.12)	−0.04 (−0.18, 0.09)
Total GWG (kg) ^b^	−0.04 (−0.55, 0.48)	−1.18 (−2.37, 0.01)	−0.37 (−0.97, 0.23)
Average Weekly GWG (kg) ^b^	0.00 (−0.03, 0.03)	−0.08 (−0.14, −0.02)	−0.03 (−0.06, 0.01)
GWG below recommendations ^a^	0.99 (0.84, 1.15)	1.46 (1.10, 1.94)	0.85 (0.60, 1.21)
GWG above recommendations ^a^	0.99 (0.89, 1.10)	0.84 (0.65, 1.09)	0.58 (0.32, 1.04)
Preterm Birth ^a^	0.74 (0.54, 1.03)	0.79 (0.40, 1.58)	1.14 (0.64, 2.03)
Caesarean Section ^a^	0.95 (0.85, 1.06)	0.80 (0.64, 1.00)	0.95 (0.72, 1.26)

Table Legend: ^a^ Binary outcome; estimates are Relative Risk of outcome (Intervention vs. Control) and 95% Confidence Interval, from a log binomial regression model, adjusted for pre-pregnancy body mass index (BMI), maternal age, smoking status, parity and the Australian Bureau of Statistics’ Socio-Economic Indexes for Areas, Index of Relative Socio-Economic Disadvantage (SEIFA IRSD) quintile (ABS 2006 LIMIT and GRoW trials, ABS 2011 OPTIMISE Trial).^b^ Continuous outcome; estimates are difference in mean (Intervention—Control) and 95% CI from linear regression model, adjusted for pre-pregnancy BMI, maternal age, smoking status, parity and SEIFA IRSD quintile.

**Table 2 nutrients-12-02314-t002:** Baseline characteristics of participants included in these analyses.

Characteristic	LIMIT Standard Care ^a^	OPTIMISE Standard Care ^a^	LIMIT Combined Groups ^b^	OPTIMISE Combined Groups ^b^	GroW Metformin ^c^	GroW Placebo ^c^
Overall Numbers	*n* = 870	*n* = 294	*n* = 1767	*n* = 585	*n* = 195	*n* = 184
Study Centre						
WCH	388 (44.60)	294 (100.00)	792 (44.82)	585 (100.00)	86 (44.10)	83 (45.11)
LMH	212 (24.37)		439 (24.84)		90 (46.15)	86 (46.74)
FMC	270 (31.03)		536 (30.33)		19 (9.74)	15 (8.15)
BMI: Median (IQR)	30.9 (27.6, 35.4)	22.2 (20.9, 23.5)	31.0 (27.8, 35.8)	22.2 (20.9, 23.6)	32.60 (28.50, 37.70)	32.27 (29.04, 36.80)
BMI Category: N (%)						
18.5–24.9		294 (100.00)		585 (100.00)	65 (33.33)	57 (30.98)
25.0–29.9	386 (44.37)		752 (42.56)		55 (28.21)	61 (33.15)
30.0–34.9	247 (28.39)		507 (28.69)		45 (23.08)	35 (19.02)
35.0–39.9	137 (15.75)		317 (17.94)		30 (15.38)	31 (16.85)
≥40.0	100 (11.49)		191 (10.81)			
Parity: N (%)					68 (34.87)	65 (35.33)
0	351 (40.34)	171 (58.16)	717 (40.58)	343 (58.63)	127 (65.13)	119 (64.67)
1+	519 (59.66)	123 (41.84)	1050 (59.42)	242 (41.37)		
Smoking N (%)					156 (84.78)	181 (92.82)
Nonsmoker	755 (86.78)	280 (95.24)	1528 (86.47)	562 (96.07)	26 (14.13)	13 (6.67)
Smoker	98 (11.26)	12 (4.08)	210 (11.88)	21 (3.59)	2 (1.09)	1 (0.51)
Missing	17 (1.95)	2 (0.68)	29 (1.64)	2 (0.34)	30.18 (5.73)	30.12 (5.47)
Age at Trial Entry: Mean (SD)	29.57 (5.41)	31.51 (4.91)	29.50 (5.41)	31.62 (4.68)		
IRSD Quintile: N (%)					55 (28.21)	66 (35.87)
Q1 (most disadvantaged)	250 (28.74)	57 (19.39)	529 (29.94)	99 (16.92)	58 (29.74)	51 (27.72)
Q2	214 (24.60)	83 (28.23)	432 (24.45)	157 (26.84)	22 (11.28)	23 (12.50)
Q3	131 (15.06)	30 (10.20)	270 (15.28)	76 (12.99)	42 (21.54)	32 (17.39)
Q4	141 (16.21)	75 (25.51)	264 (14.94)	147 (25.13)	18 (9.23)	12 (6.52)
Q5 (least disadvantaged)	134 (15.40)	49 (16.67)	271 (15.34)	106 (18.12)		

^a^ LIMIT and OPTIMISE Standard Care groups used for analysis relating to Question 1 (Is GWG modifiable through antenatal diet and physical activity?) and Question 3 (Are women with overweight/obesity at particular risk of excess GWG?). ^b^ LIMIT and OPTIMISE combined intervention and control groups used for analysis relating to Question 2 (Does optimal GWG vary by pre-pregnancy BMI category?). ^c^ GRoW Metformin and Placebo groups used for analysis relating to Question 4 (Is the association between GWG and adverse pregnancy outcomes causal?).

**Table 3 nutrients-12-02314-t003:** Relationship between diet/physical activity and gestational weight gain (GWG) in Standard Care groups from LIMIT and OPTIMISE studies.

Characteristic	LIMIT Estimate (95% CI) ^a^	LIMIT *p* Value	OPTIMISE Estimate (95% CI) ^b^	OPTMISE *p* Value
Energy (kJ): +100 kJ		0.142 *		0.403 *
BL ^c^	0.01 (−0.02, 0.03)	0.528	−0.00 (−0.02, 0.02)	0.645
28 w	−0.03 (−0.05, −0.00)	0.020	0.02 (−0.01, 0.05)	0.121
36 w	0.02 (−0.01, 0.04)	0.170	−0.02 (−0.05, 0.01)	0.185
Carbohydrate (g): +10 g		0.130 *		0.433 *
BL	0.01 (−0.06, 0.08)	0.755	−0.05 (−0.13, 0.03)	0.237
28 w	−0.08 (−0.14, −0.01)	0.029	0.05 (−0.03, 0.14)	0.233
36 w	0.06 (−0.00, 0.13)	0.070	0.02 (−0.07, 0.10)	0.696
Fiber (g): +10 g		0.360 *		0.506 *
BL	0.44 (−0.06, 0.95)	0.085	−0.26 (−0.75, 0.23)	0.303
28 w	−0.19 (−0.74, 0.36)	0.504	0.36 (−0.20, 0.92)	0.209
36 w	−0.04 (−0.59, 0.52)	0.899	−0.21 (−0.83, 0.40)	0.495
Total Fat (g): +10 g		0.637 *		0.096 *
BL	0.00 (−0.24, 0.25)	0.982	0.07 (−0.14, 0.27)	0.522
28 w	−0.17 (−0.44, 0.10)	0.225	0.19 (−0.07, 0.46)	0.151
36 w	0.09 (−0.18, 0.36)	0.515	−0.35 (−0.62, −0.08)	0.012
Protein (g): +10 g		0.293 *		0.053 *
BL	0.08 (−0.12, 0.28)	0.440	−0.03 (−0.23, 0.16)	0.752
28 w	−0.22 (−0.44, 0.00)	0.054	0.25 (0.01, 0.48)	0.038
36 w	0.09 (−0.12, 0.30)	0.389	−0.31 (−0.55, −0.07)	0.011
Sugars (g): +10 g		0.165 *		0.370 *
BL	0.08 (−0.03, 0.20)	0.166	−0.02 (−0.14, 0.11)	0.802
28 w	−0.10 (−0.21, 0.01)	0.088	0.13 (−0.02, 0.29)	0.091
36 w	0.08 (−0.03, 0.19)	0.167	−0.06 (−0.19, 0.08)	0.400
HEI: +10 units		0.862 *		0.350 *
BL	0.36 (−0.49, 1.22)	0.403	−0.31 (−1.11, 0.49)	0.451
28 w	−0.04 (−0.83, 0.76)	0.926	−0.29 (−1.03, 0.45)	0.447
36 w	−0.14 (−0.93, 0.65)	0.728	0.57 (−0.10, 1.23)	0.094
METs: +1000		0.944 *		0.721 *
BL	−0.02 (−0.17, 0.12)	0.760	0.00 (−0.14, 0.14)	0.986
28 w	0.00 (−0.16, 0.16)	0.996	−0.04 (−0.20, 0.13)	0.669
36 w	−0.03 (−0.18, 0.12)	0.704	−0.04 (−0.18, 0.09)	0.549

* denotes *p* value for global test of any association between dietary/physical activity, across all three time points and total GWG. ^a^ Estimate is the difference in mean GWG (95% CI) corresponding to the specified increase in the predictor variable (dietary intake/physical activity) at each time point, in participants from the control (Standard Care) group of the LIMIT trial. ^b^ Estimate is the difference in mean GWG (95% CI) corresponding to the specified increase in the predictor variable (dietary intake/physical activity) at each time point, in participants from the control (Standard Care) group of the OPTIMISE trial. ^c^ BL = baseline (trial entry); 28 w = 28 weeks; 36 w = 36 weeks. All estimates are derived from linear regression models, adjusted for BMI (continuous), maternal age at trial entry, parity (0 vs. 1+), smoking status and IRSD quintile.

**Table 4 nutrients-12-02314-t004:** Excess GWG by BMI category and Distance from BMI Category Cutoff.

Excess GWG	Proportion ^a^	RR (95% CI) ^b^	*p* Value
BMI Category:			0.032 *
BMI 18.5–24.9	0.18 (0.00, 0.37)	1.17 (0.96, 1.42)	0.117
BMI 25.0–29.9	0.38 (0.32, 0.43)	0.88 (0.81, 0.96)	0.004
BMI ≥ 30.0	0.46 (0.42, 0.51)	0.94 (0.92, 0.97)	<0.001

^a^ Estimated overall proportion with excess GWG within each BMI category. ^b^ Estimated RR of excess GWG for each kg/m^2^ over the BMI category cutoff. * Denotes *p* value for BMI Category by Distance from Cutoff interaction (does the effect of distance from category cutoff differ by BMI category).

**Table 5 nutrients-12-02314-t005:** Mean birth weight z-score by GWG and BMI.

Outcome	Estimate (95% CI) at GWG = 10 kg ^a^	Estimate (95% CI): Effect of 1 kg Increase in GWG ^b^	*p* Value
Birth weight z-score			0.170 *
BMI 20 kg/m^2^	−0.08 (−0.15, −0.01)	0.06 (0.05, 0.08)	<0.001
BMI 25 kg/m^2^	0.11 (0.06, 0.16)	0.06 (0.05, 0.07)	<0.001
BMI 30 kg/m^2^	0.31 (0.27, 0.35)	0.05 (0.05, 0.06)	<0.001
BMI 35 kg/m^2^	0.50 (0.45, 0.56)	0.05 (0.04, 0.06)	<0.001

^a^ Estimated mean birth weight z-score corresponding to a GWG of 10 kg, at each BMI value. ^b^ Estimated difference in mean birth weight z-score corresponding to a 1 kg higher GWG, at each BMI value. * Test of interaction between BMI and GWG.

**Table 6 nutrients-12-02314-t006:** Results of mediation analyses investigating effect of intervention and GWG on pregnancy outcomes.

Outcome	Type of Effect ^a^	Estimate (95% CI)	*p* Value
Infant large for gestational age (LGA) ^b^	Total Effect	0.97 (0.55, 1.71)	0.907
	Direct Effect	1.11 (0.62, 1.99)	0.717
	Indirect Effect	0.87 (0.73, 1.03)	0.103
Caesarean Section ^b^	Total Effect	0.56 (0.36, 0.86)	0.008
	Direct Effect	0.57 (0.37, 0.88)	0.011
	Indirect Effect	0.99 (0.94, 1.04)	0.613
Birth weight z-score ^c^	Total Effect	−0.01 (−0.21, 0.19)	0.928
	Direct Effect	0.05 (−0.14, 0.25)	0.587
	Indirect Effect	−0.06 (−0.14, 0.01)	0.090

^a^ Total Effect: Overall effect of the intervention on the outcome Direct Effect: effect of the intervention on the outcome that does not occur through effects on GWG (i.e., operating through different causal pathways) Indirect Effect: effect of the intervention on the outcome that occurs via the mediating effect of GWG. ^b^ Outcome model used logistic regression and mediator model used linear regression, adjusted for pre-pregnancy BMI, age, parity, smoking status and IRSD quintile. ^c^ Both outcome and mediator models used linear regression, adjusted for pre-pregnancy BMI, age, parity, smoking status and IRSD quintile.

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
