# Peer review of "Gestational Weight Gain—Re-Examining the Current Paradigm"

_nutrients, 2020, doi:10.3390/nu12082314_

Round 1

Reviewer 1 Report

GENERAL COMMENT

The authors aimed to investigate three specific assumptions related to the gestational weight gain (GWG) paradigm, namely 1) the modification of GWG through diet and physical activity, 2) GWG and risk of excessive GWG vary according to pre-pregnancy BMI and 3) the associations between GWG and adverse outcomes are causal. The authors used data from three RCTs conducted by their team, all using the same diet and lifestyle interventions and enrolling women from the same population. Overall, the authors found little evidence that GWG is modifiable through diet and physical activity. They were also not able to show the causality between GWG and adverse pregnancy outcomes. This paper is very interesting and raises important concerns regarding the way GWG is viewed and investigated by the research community.

MAJOR COMMENTS

  1. I would suggest adding a section regarding the limits and strengths of your analyses.
  2. Could the authors specify why they specifically chose LGA, Caesarean section and birth weight z-score as the three outcomes associated with GWG? Is it because evidence is too poor regarding other important adverse outcomes (e.g. gestational diabetes and preeclampsia. Maybe adding a sentence or two on why those outcomes were chosen and others were not.
  3. Why use only the GroW cohort to investigate the relationship between GWG and adverse pregnancy outcomes? I did not find the explanation for that decision anywhere in the manuscript. Why were the other two cohorts not included?
  4. In the Discussion section, it would be appropriate refer to the literature more often, instead of papers from the authors' own research team.
  5. There is reference to two figures (Figures 1 & 2) in the manuscript, but I did not find any figures except for the ones in the supplementary files. Where are those figures?

MINOR COMMENTS

  1. Abstract (line 15). The word ‘three’ is repeated.
  2. Abstract (line 17). ‘[…] pregnancy outcomes and with appropriate regression models, we investigated […]’
  3. I would suggest using the term ‘excessive gestational weight gain’ instead of ‘high weight gain’.
  4. Table 2. Could the authors specify what they mean by ‘Combined groups’? Is it the combination of standard care + intervention groups?
  5. Supplemental Figure 1. Not sure to understand those figures. Please add some explanations in the Results section and make the legend clearer.
  6. Discussion (line 264). ‘Our analyses used […]’

Author Response

Reviewer 1

  1. I would suggest adding a section regarding the limits and strengths of your analyses.

Thank you for this suggestion. A paragraph has been added commencing Line 346:

‘Strengths of these analyses are that the data presented here were prospectively collected from women participating in three large robust randomized trials [17-19] from the same population over a nine year period, where BMI was consistently measured by trained research staff. Outcome data were collected by trained research staff blinded to treatment allocation within the respective studies. Limitations of this work reflect the recruitment of women of predominantly Caucasian ethnicity, however this is reflective of the broader South Australian population demographic distribution. Furthermore, there is an over representation of women of high socio-economic disadvantage, with up to 75% of women declining participation. These factors may limit the external generalizability of our findings, and our methods and findings warrant replication in other available birth cohorts.’

  1. Could the authors specify why they specifically chose LGA, Caesarean section and birth weight z-score as the three outcomes associated with GWG? Is it because evidence is too poor regarding other important adverse outcomes (e.g. gestational diabetes and preeclampsia. Maybe adding a sentence or two on why those outcomes were chosen and others were not.

This has been clarified in Section 2.3, commencing Line 167:

“To investigate whether the association between GWG and adverse pregnancy outcomes is causal, such that interventions that affect GWG could thereby affect outcomes, we considered three outcomes strongly associated with ‘excess’ GWG: LGA, Caesarean section, and birth weight z-score. Data from the GRoW study were used for this analysis, as (unlike LIMIT and OPTIMISE) there was some weak evidence for an intervention effect on GWG, and on caesarean section, thus enabling investigation of the causal question by testing whether changes in GWG caused change in outcomes. As above, LGA and birth weight z-score are the outcomes for which evidence of an association with GWG is strongest; caesarean section was added as another outcome commonly associated with GWG, and for which there was some evidence of an intervention effect.[25] Other outcomes reportedly associated with ‘excess’ GWG including preterm birth and pre-eclampsia were not investigated as the number of events were too small to provide adequate statistical power, and/or there was a high risk of bias due to data not being missing at random. ….”

  1. Why use only the GroW cohort to investigate the relationship between GWG and adverse pregnancy outcomes? I did not find the explanation for that decision anywhere in the manuscript. Why were the other two cohorts not included?

As outlined in the manuscript, commencing Line 166:

“Data from the GRoW study were used for this analysis, as (unlike LIMIT and OPTIMISE) there was some weak evidence for an intervention effect on GWG and on caesarean section, thus enabling investigation of the causal question by testing whether changes in GWG caused change in outcomes.”

  1. In the Discussion section, it would be appropriate refer to the literature more often, instead of papers from the authors' own research team.

We agree the importance of representing the available literature in the discussion of any research findings. However, our research group has made significant contributions to this field internationally, particularly with regards to the breadth the findings generated from our group. Furthermore, this manuscript is the first to challenge the current and widely held paradigms around the clinical focus on gestational weight gain and the relationship with clinical pregnancy outcomes. We have completed a comprehensive literature search and have been unable to find any publications similar to ours.

  1. There is reference to two figures (Figures 1 & 2) in the manuscript, but I did not find any figures except for the ones in the supplementary files. Where are those figures?

Figures added – lines 233 and 237. Explanatory text for figure 2 added at line 220-221

MINOR COMMENTS

  1. Abstract (line 15). The word ‘three’ is repeated.

The repeated word has been removed.

  1. Abstract (line 17). ‘[…] pregnancy outcomes and with appropriate regression models, we investigated […]’

This has been edited as suggested.

  1. I would suggest using the term ‘excessive gestational weight gain’ instead of ‘high weight gain’.

We are intentionally using the word ‘high’ and not ‘excess’ as excess in this context is defined as ‘in excess of the IOM recommendations for categories of pre-pregnancy BMI’. In this paper we argue that the BMI categories used to define GWG are inadequate, and that women with a pre-pregnancy BMI close to the lower boundary of their BMI category are more likely to gain ‘excess’ weight gain. We also argue that GWG appears to be pre-determined by pre-pregnancy BMI. We acknowledge that some women do have high GWG and this is associated with greater risk of adverse pregnancy and birth outcomes.

  1. Table 2. Could the authors specify what they mean by ‘Combined groups’? Is it the combination of standard care + intervention groups?

Yes combined groups include intervention and control groups. Additional information has been added to table footnote ‘b’ on line 189.

  1. Supplemental Figure 1. Not sure to understand those figures. Please add some explanations in the Results section and make the legend clearer.

The following has been add at line 261-264:

‘Supplementary figures 1-3 illustrate the relationship between pre-pregnancy BMI and GWG (standardized as number and standard deviation (SD) from the mean) in the control groups from all three studies along with a) proportions of LGA infants, b) proportion of caesarean deliveries, and c) birth weight z-score. The proportion of women with excess GWG decreases as BMI increases, whilst proportion of LGA, caesarean birth and birth weight z-score increase.’

This statement has been added to the legend accompanying the figures.

  1. Discussion (line 264). ‘Our analyses used […]’

We have considered this suggestion but prefer the word ‘utilized’.

Reviewer 2 Report

Manuscript “Gestational weight gain – re-examining the current paradigm” (Nutrients 883802).

The authors have extensive experience in research on nutrition in pregnancy. On this occasion, the aim of the review was to investigate the underlying assumptions of the current GWG paradigm, using data from their suite of harmonized RCTs (LIMIT, GROW and OPTIMISE trials).

Comments and Suggestions for Authors:

The manuscript is an interesting review, but requires some considerations.

The biggest limitation of the review is that the participants were recruited from the same population. Predominantly white Caucasian, with less than half of women from areas of high social disadvantage and furthermore, 75% of eligible women declined participation (OPTIMISE trial). These factors may limit the external validity and generalisability of the findings to other patient populations.

Tables. The subsections of each variable are preceded by hyphens (-) that make their interpretation difficult, especially when it comes to numerical variables, for example: - 18.5-24.9. If they were not used it would improve their visualization.

Page 2. Line 209. What is Figure 1?

Page 4. Line 277. What is Figure 2?

Discussion. According to the authors, the BMI should always be considered as a continuous variable and not using categories?

References should be thoroughly revised to conform to uniform and appropriate standards for the journal Nutrients.

Author Response

  1. The biggest limitation of the review is that the participants were recruited from the same population. Predominantly white Caucasian, with less than half of women from areas of high social disadvantage and furthermore, 75% of eligible women declined participation (OPTIMISE trial). These factors may limit the external validity and generalisability of the findings to other patient populations.

The following has been added to line 334:

‘Strengths of these analyses are that the data presented here were prospectively collected from women participating in three large robust randomized trials [17-19] from the same population over a nine year period, where BMI was consistently measured by trained research staff. Outcome data were collected by trained research staff blinded to treatment allocation within the respective studies. Limitations of this work reflect the recruitment of women of predominantly Caucasian ethnicity, however this is reflective of the broader South Australian population demographic distribution. Furthermore, there is an over representation of women of high socio-economic disadvantage, with up to 75% of women declining participation. These factors may limit the external generalizability of our findings, and our methods and findings warrant replication in other available birth cohorts.’

  1. Tables. The subsections of each variable are preceded by hyphens (-) that make their interpretation difficult, especially when it comes to numerical variables, for example: - 18.5-24.9. If they were not used it would improve their visualization.

 These hyphens have been removed.

  1. Page 2. Line 209. What is Figure 1?

 Figure 1 has been added. Line 234

  1. Page 4. Line 277. What is Figure 2?

Figure 2 has been added. Line 238.

  1. Discussion. According to the authors, the BMI should always be considered as a continuous variable and not using categories?

I am not sure what the reviewer is suggesting here.

  1. References should be thoroughly revised to conform to uniform and appropriate standards for the journal Nutrients.

Format of references has been edited.